# Atrial Fibrillation and Dementia: Pathophysiological Mechanisms and Clinical Implications

**DOI:** 10.3390/biom14040455

**Published:** 2024-04-08

**Authors:** Dimitrios Varrias, Tinatin Saralidze, Pawel Borkowski, Sumant Pargaonkar, Michail Spanos, George Bazoukis, Damianos Kokkinidis

**Affiliations:** 1Department of Medicine, Jacobi Medical Center, Albert Einstein College of Medicine, Bronx, NY 10461, USApawel.borkowski.md@gmail.com (P.B.); sumant.pargaonkar@gmail.com (S.P.); 2Cardiovascular Research Center, Massachusetts General Hospital, Harvard Medical School, Boston, MA 02114, USA; mspanos1@mgh.harvard.edu; 3School of Medicine, European University Cyprus, 2417 Nicosia, Cyprus; 4Section of Cardiovascular Medicine, Yale University, New Haven, CT 06520, USA

**Keywords:** dementia, atrial fibrillation, Alzheimer’s, stroke, bleeding, anticoagulation

## Abstract

Numerous longitudinal studies suggest a strong association between cardiovascular risk factors and cognitive impairment. Individuals with atrial fibrillation are at higher risk of dementia and cognitive dysfunction, as atrial fibrillation increases the risk of cerebral hypoperfusion, inflammation, and stroke. The lack of comprehensive understanding of the observed association and the complex relationship between these two diseases makes it very hard to provide robust guidelines on therapeutic indications. With this review, we attempt to shed some light on how atrial fibrillation is related to dementia, what we know regarding preventive interventions, and how we could move forward in managing those very frequently overlapping conditions.

## 1. Introduction

Numerous longitudinal studies suggest a strong association between cardiovascular risk factors and cognitive impairment. Patients with atrial fibrillation (AF) are at higher risk of dementia and cognitive dysfunction as AF increases the risk of cerebral hypoperfusion, inflammation, and stroke [1]. However, the existing literature still lacks a thorough analysis of the observed association and complex relationship between these two diseases. In addition, no official guidelines exist to provide clear indications for treatment based on prospective intervention studies. This review will provide a comprehensive and detailed overview of the topic.

## 2. Epidemiology of AF and Dementia

AF is the most prevalent arrhythmia in adults, with around 37.5 million cases; at the same time, dementia accounts for a massive burden for the healthcare system, with an estimated prevalence of approximately 55 million cases. With the population ageing worldwide, epidemiological predictions foresee increased prevalence of both disorders as well as associated morbidity and mortality [2]. Saglietto et al. reported atrial fibrillation and dementia epidemiological data as analyzed from 1990 to 2019 [3]. The overall incidence rate of AF and dementia was 7–9 folds higher in higher sociodemographic index (SDI) countries compared to lower SDI countries. Remarkably, there is a significant overlap in the population affected by the two diseases, with Liu et al. reporting that AF was independently associated with dementia (HR = 1.34, 95% CI: 1.24–1.44) [4]. Unfortunately, the causal association is still being investigated [5].

## 3. Challenges in Associations

There are many potential links between AF and dementia. Firstly, a well-known risk factor for dementia is stroke, which is one of the most common complications of AF. Second, a decline in cardiac output secondary to AF leads to chronic cerebral hypoperfusion, which in turn causes central nervous system dysfunction. Although associating the two conditions appears obvious, significant gaps in the mechanistic etiology of AF-related cognitive consequences exist beyond aging and stroke. The shared etiology due to overlapping risk factors (including advanced age, hypertension, heart failure, diabetes, hyperlipidemia, sleep apnea, coronary artery disease, chronic kidney disease, obesity, and physical inactivity) as well as therapeutic options (DOACS) [2] acts as a significant confounder in existing observational studies. Furthermore, causal associations with dementia may be complicated to prove due to the slowly progressive nature of the disease. Still, it also has many different sub-categories (vascular, Alzheimer’s, etc.) [6]. Recently, several longitudinal studies have suggested that AF correlates with all causes of dementia but also with Alzheimer’s disease [6,7]. However, observational studies might be confounded by potential biases and reverse causation [8]. On the other hand, solid evidence from numerous prospective studies has shown AF to be associated with cognitive impairment, cognitive decline, and all causes of dementia [7,9,10,11,12,13,14]. However, most studies examined the relationship with all-cause dementia without separate investigation for vascular dementia and Alzheimer’s disease [10,11,12,13,14]. Although there is evidence that AF correlates with Alzheimer’s, there is no suggested mechanical relationship between the two, with studies concluding that it is likely related to confounding bias due to shared comorbidities [6]. Objective quantification of dementia can be challenging but vital when it comes to studies regarding associations. MRI is the gold standard as it can depict structural changes (lacunar infarcts, atrophy, and amyloid depositions) [15]. Computed tomography, despite being more cost effective, does not provide optimal results when it comes to differentiating types of dementia [16]. Recently, there has been a lot of progress in the field of neuro- imaging, with advanced metabolic and functional modalities. Identifying pathology before the radiographic evidence in T2 weighted MRI is a challenge, but single-photon emission computed tomography, fluorodeoxyglucose–positron emission tomography and blood oxygenation level–dependent MRI are state-of-the art and promising modalities [17].

## 4. Evidence of Association 

Several well-established population-based longitudinal studies have demonstrated a higher risk of cognitive decline or dementia associated with AF. Nevertheless, most of the evidence comes from the data on the elderly population. Hence, the association might be related to underlying systemic vascular disease and the increased prevalence of both disorders with decreased age. Therefore, it is crucial to investigate whether the link arises from a common pathophysiological mechanism or whether AF plays a role in a causal pathway. Interestingly, in the Whitehall study [14], more than 10,000 patients (aged 45–69 years) who completed cognitive tests four times over 15 years were recruited. Compared to AF-free participants, patients with longstanding AF (5, 10, or 15 years) experienced a steeper curve of cognitive decline and a higher risk of dementia after adjusting confounders (sociodemographic, behavioral, and chronic diseases) [HR: 1.87; 95% CI: 1.37, 2.55]. Hence, in those with early AF onset, a more extended exposure period led to more significant neuronal injury and loss, possibly due to an enhancing effect of the degenerative disease on vascular changes. In addition, a study showed that even adults with incident AF at age 50–55 had accelerated cognitive decline [14]. Atrial fibrillation was associated with an increased risk of dementia (hazard ratio, 1.23; 95% confidence interval, 1.04–1.45), even after adjusting for cardiovascular risk factors, including ischemic stroke [18]. This association was strongest for younger patients who consistently had a longer duration of AF. A systematic review and meta-analysis by Kwork [19] identified 15 relevant studies, including 46,637 participants with a mean age of 71.7 years. Fourteen studies showed a significant increase in the risk of dementia associated with AF (OR 2.0, 95% CI 1.4 to 2.7, *p* < 0.0001), with substantial heterogeneity (I(2) = 75%). After stratification by participants, the association was significant in studies focusing only on stroke patients (7 studies, OR 2.4, 95% CI 1.7 to 3.5, *p* < 0.001, I(2) = 10%), and of marginal significance in broader populations (7 studies, OR 1.6, 95% CI 1.0 to 2.7, *p* = 0.05, I(2) = 87%). Santangeli et al. comprised eight prospective studies including more than 77,000 patients, of whom 11,700 (17%) had AF [20,21,22,23,24,25,26,27,28]. The authors investigated whether there is an association between AF and dementia in patients with normal cognitive function at baseline who did not suffer an acute stroke. After adjusting for confounders, an independent risk of developing dementia was demonstrated in AF patients with an HR of 1.4 (95% CI 1.2–1.7) [7]. An even more extensive systematic review published by Kalantarian assessed the association of AF with cognitive decline, including prospective and non-prospective studies, mainly using MMSE and DSM-III or IV criteria. What was found was that the risk for cognitive impairment was more than doubled in AF patients with a history of stroke [RR 2.70 (95% CI 1.82–4.00)]. Additionally, there was a markedly increased risk for cognitive decline regardless of stroke history [RR 1.40 (95% CI 1.19–1.64)] [29]. There is limited data available regarding the prevalence of dementia within various treatment option populations for AF. The observed inconsistencies between studies that establish associations and those that don’t could be due to the variations in the methodology used across studies, a lack of rigorous adjustment for potential confounder variables, and having highly selected populations. For instance, certain studies have small sample sizes or short follow-up periods. In addition, different approaches and age ranges were used to assess AF and dementia across the studies. In summary, current evidence from systematic reviews and population-based longitudinal studies suggests an independent association of dementia and cognitive decline with AF, especially among the age group of 64–74.

## 5. Proposed Mechanistic Associations between AF and Cognitive Dysfunction

Cerebral infarction, AF-related cerebral hypoperfusion, microbleeds, inflammation, brain atrophy, atherosclerosis, and others (Figure 1).

### 5.1. Silent Cerebral Infarcts

There have been many studies showing that silent cerebral infarcts are associated with dementia and stroke. In a meta-analysis, AF was associated with a 2.6-fold increased risk of silent cerebral infarcts [30]. Compared with controls in sinus rhythm, patients with AF had much lower cognitive function in an observational study that assessed patients with silent infarcts (validated by MRI) [31]. In studies using systematic brain imaging, 15% to 50% of patients with AF have a brain infarct [32]. What is more remarkable is that silent infarcts at the time of diagnosis of AF are up to five times more common compared to symptomatic infarcts [33]. As highlighted by Healy et al., frequently silent infarcts are miss-characterized as “innocent, underestimating their clinical significance and the risk of developing into fatal strokes [34]. Although this may often be regarded as a benign incidental finding without significant neurologic deficits at the time, it is associated with both concurrent cognitive performance and risk for cognitive decline [35,36]. The prospective SWISS-AF showed that the infarct size is crucial to delayed outcomes. It was described that only the large embolic silent noncortical or cortical infarcts were associated with lower cognitive scores in patients with AF, but not the smaller size infarcts [37].

### 5.2. Cerebral Microbleeds

Microbleeds in key brain locations (lobar locations) have been reported to be associated with poor cognition. What is remarkable is that even after adjusting for vascular risk factors and imaging markers of CSVD, this relationship persists [38]. On the other hand, in a large cohort of patients with AF, there was no association between cerebral microbleeds and cognitive dysfunction [38]. Whether microbleeds are directly causally related to cognitive impairment or they are bound to our therapeutic management of AF is still a mystery. There is a chance that microbleeds attributable to OAC pose an explanation for cognitive decline in patients with AF. A prospective MRI study found that only warfarin (not direct oral anticoagulants or antiplatelet agents) was associated with developing new microbleeds at one year in patients with AF. Still, we need larger cohorts to exclude that pathophysiologic mechanism from the list [22].

### 5.3. Cerebral Hypoperfusion

Transient or chronic cerebral hypoperfusion has been implicated in the development of dementia [39]. AF decreases cardiac output due to atrioventricular desynchrony, resulting in less stroke volume and lower blood pressure [7]. Another proposed mechanism is an interbeat volume variation, which may contribute to transient cerebral hypoperfusion [23]. Although cerebral auto-regulation is expected to keep blood flowing to the brain during a wide blood pressure range, several studies have noted decreased cerebral perfusion in patients suffering from AF [40]. In 358 patients with cognitive impairment and an age greater than 65 years, excluding those with a history of transient ischemic attack, stroke, or dementia, AF was an independent predictor for dementia, increasing the risk fourfold after a mean follow-up of 10 years [41]. Relevant to the cerebral hypoperfusion hypothesis, patients with either a low or high heart rate (<50 or >90) on 24-h Holter monitoring had a 7-fold risk for dementia compared to patients with a normal ventricular rate. These results are supported by computational data highlighting that faster rates relate to a progressive decrease in cerebral perfusion and hypotensive events in the cerebral circulation [42]. Relevant to the size of silent infarcts predicting cognitive impairment, the duration of AF was independently associated with worse outcomes. In a cross-sectional study, patients with sustained AF had reduced brain flow compared to those with paroxysmal AF or sinus rhythm [24].

### 5.4. Inflammation

Inflammation enhances hypercoagulability and potentiates the formation of thrombi, increasing the risk for stroke and malfunction of cerebrovascular regulation, which has been linked to Alzheimer’s and vascular dementia [28,43]. AF is a pro-inflammatory condition [26,27]. Validating the above, studies showed that increased markers of inflammation correlate with cognitive impairment in patients with AF. Patients who developed dementia were shown to have impaired hemostatic ability, resulting in microbleeds, as discussed above. Continued microbleeds and inflammation form a vicious cycle that perpetuates, leading to a cascade of unfavorable physiologic changes. This association between hemostasis and vascular dementia is also expressed as an abundance of thrombin generation markers (D-dimer and prothrombin fragment 1 + 2) and endothelial dysfunction (von Willebrand factor and plasminogen activator inhibitor) [44]. In rodent models of Alzheimer’s disease, dabigatran (a direct thrombin inhibitor) was shown to improve pathologic changes, reduce inflammation, and slow down cognitive dysfunction even in the absence of AF [45]. Thus, inflammatory markers could correlate with the degree of atherosclerosis associated with cognitive impairment and AF [46].

### 5.5. Systemic Atherosclerosis/CHADVASC

As discussed above, atherosclerosis is strongly associated with dementia. Preclinical markers of cardiovascular disease have been linked to both an increased risk of AF and cognitive impairment [46,47,48]. Some are subclinical atherosclerosis, aortic stiffness, and intima-media thickness of the carotid artery. Similarly, there is a direct correlation between higher CHADS2 and CHA2DS2VASc scores and the risk of developing cognitive dysfunction in patients with AF [49]. However, the number of studies on the relationship between those scores and cognitive impairment is limited [16,17]. Chou et al. investigated the correlation between vascular dementia/Alzheimer’s disease and CHADS2 score in the Taiwan cohort, [16] concluding that the CHADS2 score is a valuable predictor. In another study using the Taiwan AF cohort of 332.665 patients with AF, Liao et al. showed that CHADS2 and CHA2DS2-VASc were independently associated with developing dementia during a 14-year follow-up. However, the CHA2DS2-VASc score was better at predicting dementia [17], with a score of four or greater having 1.5 times more chances of developing dementia than patients with less than that [11].

### 5.6. Genetics and Biomarkers

During the last ten years, significant progress has been made toward detecting biomarkers of diagnostic and prognostic value for dementia [50]. Identifying biomarkers predicting cognitive decline in AF patients could inform screening and management strategies. Biomarkers will undoubtedly help refine risk stratification [51], but as with most novel biomarkers (lncRNA BACE1-AS, MALAT1) [52], we must balance predictive ability against practicality [50]. In a Mendelian randomization study using 93 SNPs as the instruments, Pan et al. showed an insignificant association of genetically predicted AF with the risk of Alzheimer’s disease. In detail, both a fixed-effect and a random-effect inverse-variance weighted Mendelian randomization (IVW-MR) method showed that genetically predicted AF was not associated with the risk of Alzheimer’s disease (OR = 1.002, 95% CI: 0.996–1.009, *p* = 0.47; OR = 1.002, 95% CI: 0.995–1.010, *p* = 0.52). This study concludes that genetically predicted AF had no causal effect on the risk of Alzheimer’s disease, with the findings being significant even when a sensitivity analysis was performed [6].

### 5.7. Amyloid Aβ_42_

Atrial fibrillation compromises cerebral blood flow, which results in the development of senile plaques due to the long-term deposition of Aβ_42_ [53]. What was known was that this type of cerebral amyloid deposition is associated with poor cognitive function and the development of dementia [54,55]. During the last decade, there have been tremendous advances in linking cerebral amyloid disease with low flow states, as in AF. Niwa et al. showed that atrial fibrillation leads to vascular dysfunction which aggravates the deposition of amyloid [56]. Hawkes et al. demonstrated how amyloid Aβ_2_ hinders the clearance of cerebral amyloid, creating a vicious cycle [57]. Finally, Yao et al. elaborated on the role of hypoperfusion and cerebral amyloid disease, attributing it to activation of the tau phosphorylation enzymes glycogen synthase kinase-3 beta and cyclin-dependent kinase-5, which eventually induce the accumulation of Aβ_42_.

## 6. Treatment

### 6.1. Rhythm Control

The potential impact of successful rhythm control on cognitive decline among patients with atrial fibrillation is an area of ongoing debate. Data originating from a recent study hint at the beneficial role of electric cardioversion in cerebral flow assessed by MRI [24]. Improved hemodynamics and increased brain perfusion resulting from rhythm control are hypothesized to delay cognitive decline. On the other hand, most of the existing literature reports that electric cardioversion increases the risk of cerebral microemboli. In a population of AF patients, there was no consistent evidence of an association between PVI (pulmonary vein isolation) and neurocognitive function [58]. In other studies, AF ablation has been associated with declining cognitive function and acute brain lesions [59,60,61]. However, a direct association between silent cerebral microembolism and cognitive decline is yet to be proven [62]. Findings from the EAST-AFNET 4 trial did not demonstrate a significant impact of early rhythm control on cognitive function [63]. Despite this, ongoing studies, such as the Comparison of Brain Perfusion in Rhythm Control and Rate Control of Persistent Atrial Fibrillation (NCT02633774) and AFCOG (NCT04033510), aim to investigate the role of rhythm control in preventing cognitive decline in AF. The results of these ongoing trials will inform treatment strategies to improve the cognitive outcomes of individuals with AF and potentially impact this patient population’s management [1]. Since this is an issue of vital importance, the NOR-FIB2 (Fibrosis, Inflammation, and Brain Health in Atrial Fibrillation: The Norwegian Atrial Fibrillation and Stroke Study; URL: https://www.clinicaltrials.gov; Unique identifier: NCT03816865) is ongoing and ventures to assess whether programmed direct-current cardioversion increases the risk of cognitive dysfunction and the incidence of new-onset silent cerebral infarcts. In summary, the results seem to be either neutral or negative regarding the effect of cardioversion on dementia. Still, more studies are required to investigate whether converting to sinus could reduce the risk of developing dementia in the long run.

### 6.2. Comparison between Ablation and Oral Antiarrhythmics

Compared to oral antiarrhythmics, ablations reduce paroxysmal AF episodes and prolong the duration of sinus normorhythmia, thereby improving the quality of life [64,65]. Two recent studies demonstrated that, beyond that, catheter ablation improved cognitive function [66,67]. Using the Korean NHIS database, Kim et al. [68] compared the risk of dementia between 9119 patients undergoing ablation and 17,978 patients managed using medical therapy (antiarrhythmic or rate control drugs). During a median follow-up of 52 months, ablated patients had a significantly lower incidence and a reduced risk of dementia overall (8.1 and 5.6 per 1000 person-years, respectively; HR, 0.73; 95% CI, 0.58–0.93). The authors inferred that restoring sinus rhythm, not the ablation procedure, was the critical mechanism in reducing the risk for dementia since the ablation group had longer durations of persistent sinus rhythm when compared to the medical management arm. The ongoing DIAL-F case–control study (Cognitive Impairment in Atrial Fibrillation; Unique identifier: NCT01816308) is comparing the incidence of cognitive impairment (assessed by MoCA) in two groups of patients with AF (Patients undergoing catheter ablation for AF versus those receiving antiarrhythmic drugs) [1].

### 6.3. Anticoagulation

The pathophysiological mechanisms linking atrial fibrillation with cognitive impairment may be attributed to both microembolization (silent strokes) and macroembolization (overt strokes). Anticoagulation therapy is an effective measure to prevent stroke and systemic emboli in AF patients. However, anticoagulation-associated microbleeds can worsen cognitive function, concluding that intervention RCTs are needed to assess long-term outcomes. There are incoming data that suggest that OACs are lowering the risk of AF-related dementia [69,70,71]. Kim et al. reported that OACs had a preventive effect on dementia with an HR of 0.61 among patients with incidental AF. Friberg and Rosenqvist [70] found that patients with AF who received OAC treatment had a 29% lower risk of dementia than those with AF who were not prescribed OACs (HR, 0.71; 95% CI, 0.68–0.74) in a large-scale national cohort in Sweden. That study also showed a suggested dose-dependent effect, with patients lowering their risk the longer they are on an OAC. In another study by the same authors, it was recommended that AF patients aged >65 years, irrespective of their stroke risk score, benefit from OAC in regards to lowering the risk for dementia [71]. Several clinical trials, including GIRAF [72] and CAF [73], have investigated the role of warfarin and dabigatran in preventing cognitive decline. Both trials evaluated the participants at baseline and two years after the initiation of the study and revealed no significant differences between the two medications. Despite some studies reporting a benefit, others are suggesting that anticoagulation is associated with higher risks of developing dementia in AF, possibly due to microbleeds. In AF patients receiving warfarin, a lower time-in-therapeutic range has been associated with a higher risk of dementia [21,74]. Due to frequent suboptimal control in patients receiving warfarin, there is a significant concern for microbleeds, which can, in line with this, cause chronic cerebral injuries and inflammation and eventually lead to cognitive decline [75].

### 6.4. Type of AC

Currently, there are no randomized data on the efficacy of different OACs in preventing dementia in AF patients. In patients with an indication for AC, it is not ethically acceptable to randomize patients to OAC versus placebo to assess cognitive outcomes. On the other hand, small observational studies highlight the protective effect of DOACs compared to other pharmacological options when it comes to cognitive outcomes [70,76]. A small meta-analysis of four randomized trials that compared NOACs to warfarin demonstrated that NOACs were associated with a significant risk reduction in terms of overall stroke and systemic embolism [77,78]. Two cohort studies performed in the USA also agreed with the above outcome [76,79]. While the Danish registries showed no difference in dementia between DOACs and warfarin users [80], Kim et al. [81] enrolled 52,888 new OAC users with AF (aged ≥60 years, 31,211 NOAC users, and 21,677 warfarin users) from the Korean NHIS database, suggesting statistically significant differences in outcomes. Relative to propensity-matched warfarin users, NOAC users, regardless of whether it was apixaban, dabigatran, or rivaroxaban, tended to have a lower risk of dementia (HR, 0.78; 95% CI, 0.69–0.90). Nationwide Swedish and Danish cohort studies showed neutral results, as they concluded that the incidences of dementia when comparing warfarin and NOAC were similar [70,80].

### 6.5. Anti-Inflammatory Agents

The relationship between inflammation and cognitive decline in individuals with atrial fibrillation remains an area of active research. While some evidence suggests that inflammation plays a role in the development of cognitive impairment in patients with AF, further investigation is necessary to fully understand this relationship’s nature. Specifically, additional research is required to determine the potential effectiveness of anti-inflammatory medications, such as statins, aspirin, or etanercept, in mitigating the cognitive decline associated with inflammation. Statin therapy appears to positively affect cognitive outcomes in patients with AF [70], but this will require more extensive studies to influence our current guidelines.

## 7. Future Directions and Conclusions

Understanding the relationship between atrial fibrillation and structural brain changes is a crucial area of investigation in cognitive neuroscience. However, the underlying structural changes that may contribute to this decline still need to be fully understood. In this review, we summarized the potential etiological link between AF and dementia, analyzed potential risk stratification or monitoring surrogates, and elaborated on therapeutic interventions that have shown benefit (Figure 2).

Hopefully, as more light is shed on the subject, more trials will be designed to solidify therapeutic guidelines. Ongoing and recently published studies, such as the SWISS-AF study [82], represent an essential step forward in clarifying the nature of this relationship. Rhythm control using pharmacological cardioversion, electrical cardioversion, or ablation to enhance cerebral perfusion and reduce the incidence of cognitive decline in patients with atrial fibrillation is of significant interest. Additionally, investigations into different anticoagulants and their effect on cognitive impairment, the potential benefits of anti-inflammatory medications such as statins in mitigating the impact of systemic inflammation associated with AF, and the role of genetics in determining susceptibility to cognitive decline are all promising areas for future research. Finally, as the neurodiagnostic tools continue to evolve, we hope to identify more and more links between neurophysiology and low blood flow states, even before atrophy begins to manifest on a macroscopic level. Continued collaboration between basic science and clinical research is critical in identifying novel approaches to preventing and managing cognitive decline in AF patients. The anticipated findings from these studies can impact early intervention strategies and ultimately improve patient outcomes.

## Figures and Tables

**Figure 1 biomolecules-14-00455-f001:**
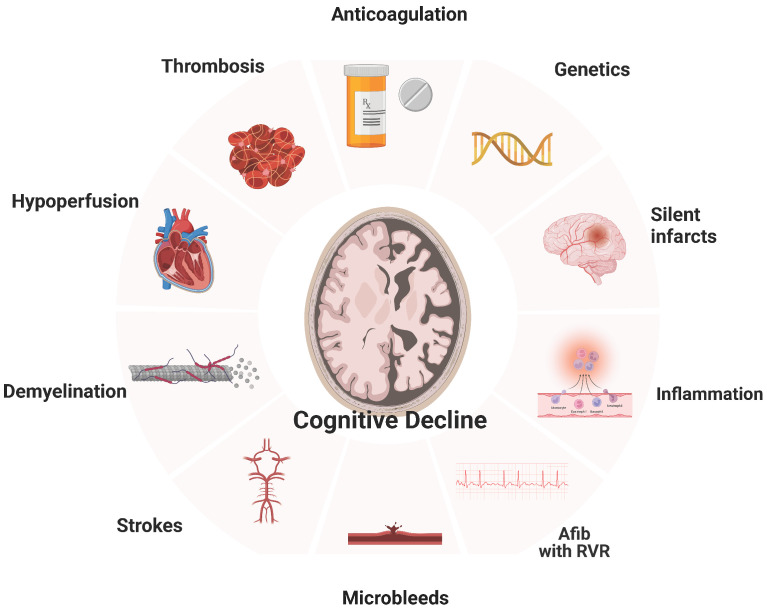
Relationship of Atrial Fibrillation with Dementia.

**Figure 2 biomolecules-14-00455-f002:**
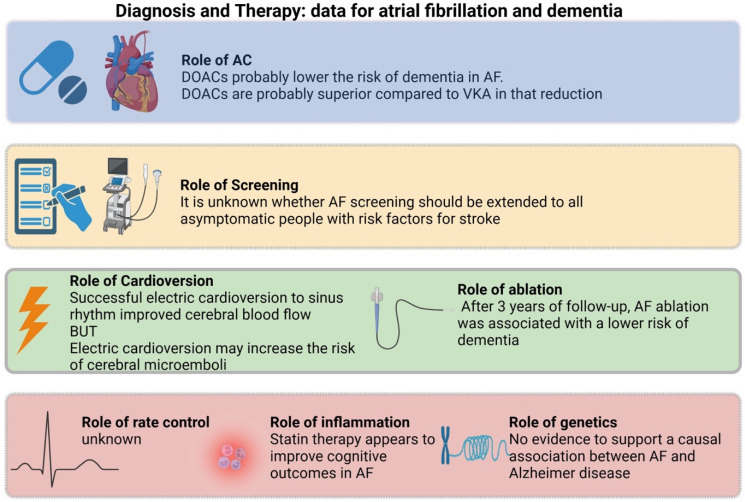
Summary of evidence regarding atrial fibrillation and dementia.

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
