# Peer review of "Atrial Fibrillation and Dementia: Pathophysiological Mechanisms and Clinical Implications"

_biomolecules, 2024, doi:10.3390/biom14040455_

Round 1

Reviewer 1 Report

Comments and Suggestions for Authors

The article has a good structure. Although the aim of it is to shed some light on how atrial fibrillation is related to dementia, it has only managed to emphasize and provide an in depth analysis of what is known in the literature. Nevertheless it will guide present and future physicians to better understand the relation between them, screen for possible risk factors and better manage both diseases.

Comments on the Quality of English Language

Row nr 32: maybe use another word for constitutes( ex: accounts for)

Row nr 33: please use ageing instead of aging

Row 37 & 38: either use higher and lower SDI for both or high and low SDI for both.

It would be interesting to describe some of the methods used to assess cognitive function.

ROW 199: could you please provide an example of such novel biomarkers, even if they are in an early stage?

Author Response

Thank you for your kind remarks. We really appreciate your time in reviewing our paper and providing valuable insights. We made the following corrections in response to your comments: 

Row nr 32: maybe use another word for constitutes( ex: accounts for): DONE

Row nr 33: please use ageing instead of aging: DONE

Row 37 & 38: either use higher and lower SDI for both or high and low SDI for both.: DONE

It would be interesting to describe some of the methods used to assess cognitive function.: Added more info on the requested topic. 

ROW 199: could you please provide an example of such novel biomarkers, even if they are in an early stage? Added specific examples of state of the art novel biomarkers that are currently being tested: long non coding RNA  BACE1-AS ( upregulated and MALAT 1 ( downregulated). 

In addition to the above, another revision was made regarding the use of English language in order to achieve the maximum score from your side. 

Best 

Reviewer 2 Report

Comments and Suggestions for Authors

In the present manuscript authors aim at shedding some light on how atrial fibrillation is related to dementia and evalutating preventive interventions.

The topic is interesting but several other similar articles have been already published on this field. Only in the last 5 years about 600 manuscripts are present on pubmed, including review and meta-analysis. Thus, I am not sure this article could add anything new. 

Author Response

Dear reviewer, 

Thank you so much for taking the time to review our recent work. 

Although we understand your concerns, we feel that this is an unfair criticism given the fact that this is an invited review for a special issue. Since our aim though is to improve, based on your comments, we tried and will continue trying to incorporate as many key advances described in the latest literature, providing additional data to this complex and nuanced topic in our current revision.  

Thank you once more for your time. 

Round 2

Reviewer 2 Report

Comments and Suggestions for Authors

Dear authors,

thank you for your reply but this is the opinion of this reviewer

Regards